# The Impact of COVID-19 Epidemic Declaration on Psychological Consequences: A Study on Active Weibo Users

**DOI:** 10.3390/ijerph17062032

**Published:** 2020-03-19

**Authors:** Sijia Li, Yilin Wang, Jia Xue, Nan Zhao, Tingshao Zhu

**Affiliations:** 1Institute of Psychology, Chinese Academy of Sciences, Beijing 100101, China; lisj@psych.ac.cn (S.L.); 1613118@mail.nankai.edu.cn (Y.W.); 2Department of Psychology, University of Chinese Academy of Sciences, Beijing 100049, China; 3Department of Psychology, Nankai University, Tianjin 300071, China; 4Factor Inwentash Faculty of Social Work, University of Toronto, Toronto M5S 1A1, Canada; jia.xue@utoronto.ca

**Keywords:** public health emergencies, word frequency analysis, mental health, emotion, cognition

## Abstract

COVID-19 (Corona Virus Disease 2019) has significantly resulted in a large number of psychological consequences. The aim of this study is to explore the impacts of COVID-19 on people’s mental health, to assist policy makers to develop actionable policies, and help clinical practitioners (e.g., social workers, psychiatrists, and psychologists) provide timely services to affected populations. We sample and analyze the Weibo posts from 17,865 active Weibo users using the approach of Online Ecological Recognition (OER) based on several machine-learning predictive models. We calculated word frequency, scores of emotional indicators (e.g., anxiety, depression, indignation, and Oxford happiness) and cognitive indicators (e.g., social risk judgment and life satisfaction) from the collected data. The sentiment analysis and the paired sample t-test were performed to examine the differences in the same group before and after the declaration of COVID-19 on 20 January, 2020. The results showed that negative emotions (e.g., anxiety, depression and indignation) and sensitivity to social risks increased, while the scores of positive emotions (e.g., Oxford happiness) and life satisfaction decreased. People were concerned more about their health and family, while less about leisure and friends. The results contribute to the knowledge gaps of short-term individual changes in psychological conditions after the outbreak. It may provide references for policy makers to plan and fight against COVID-19 effectively by improving stability of popular feelings and urgently prepare clinical practitioners to deliver corresponding therapy foundations for the risk groups and affected people.

## 1. Introduction

COVID-19 (Corona Virus Disease 2019) is a highly infectious disease with a long incubation period which was caused by Sars-Cov-2 (Severe Acute Respiratory Syndrome Coronavirus 2) [1]. The number of COVID-19 patients increased dramatically due to hundreds of millions of people traveling during the Spring Festival period. The severity of COVID-19 had been underestimated until the National Health Commission classified it as a B type infectious disease officially and took actions to fight against this disease on 20 January, 2020. Ever since then, epidemic prevention was comprehensively upgraded and marked the real beginning of universal concern, indicating widespread impacts.

The uncertainty and low predictability of COVID-19 not only threaten people’s physical health, but also affect people’s mental health, especially in terms of emotions and cognition, as many theories indicate. According to Behavioral Immune System (BIS) theory [2], people are likely to develop negative emotions (e.g., aversion, anxiety, etc.) [3,4] and negative cognitive assessment [5,6] for self-protection. Faced with potential disease threat, people tend to develop avoidant behaviors (e.g., avoid contact with people who have pneumonia-like symptoms) [7] and obey social norms strictly (e.g., conformity) [8]. According to stress theory [9] and perceived risk theory [10], public health emergencies trigger more negative emotions and affect cognitive assessment as well. These negative emotions keep people away from potential pathogens when it refers to the disease. However, long-term negative emotions may reduce the immune function of people and destroy the balance of their normal physiological mechanisms [11]. Meanwhile, individuals may overreact to any disease in case of less appropriate guidance from authorities, which may result in excessively avoidant behaviors and blind conformity [8]. Therefore, it is essential to understand the potential psychological changes caused by COVID-19 in a timely manner. Since psychological changes caused by public health emergencies can be reflected directly in emotions and cognition [3,4,5,6], we can monitor psychological changes in time through emotional (e.g., negative emotions and positive emotions) and cognitive indicators (e.g., social risk judgment and life satisfaction).

The emotions and cognition are usually measured by retrospective questionnaires, such as Oxford Happiness Inventory (OHI) [12], Symptom Checklist 90 (SCL-90) [13], Satisfaction with Life Scale (SWLS) [14], and Likert Type Attitude Scale [15,16]. However, at the time of the COVID-19 outbreak in China, it was very difficult to conduct a traditional paper survey in the affected areas; online surveys rely on the cooperation of participants, and it is difficult to meet the requirements in time, and even brings extra burdens for participants. Since we did not know the time of COVID-19 declaration, it was impossible to measure people’s emotions and cognition by a traditional survey in advance. There may be a certain deviation when requiring people to recall their mental state a week or more ago. Weibo data is emerging as a key online medium and data source for researchers to understand this social problem in a non-invasive way. Sina Weibo is a leading Chinese Online Social Networks (OSN) with more than 462 million active daily users in 2019. These users use Weibo functions (e.g., reply, @function) to interact with each other, forming rich user behavior data.

The aim of this study is to explore the impacts of public health emergency COVID-19 on people’s mental health, to assist policy makers to develop actionable policies, and help clinical practitioners (e.g., social workers, psychiatrists, and psychologists) provide services to affected populations in time.

## 2. Materials and Methods

### 2.1. Participants and Data Collection

The samples in this study were from the original Weibo data pool [17]. The data pool contained more than 1.16 million active Weibo users. Weibo is a popular platform to share and discuss individual information and life activities, as well as celebrity news in China [18].

The retrieved data included (1) user’s profile information, (2) network behaviors, and (3) Weibo messages. Privacy was strictly protected during the procedure, referring to the ethical principles [19]. We have obtained the Ethical Committee’s approval and the ethic code is H15009.

The following inclusion criteria were employed to select active Weibo users from the data pool. First, they had published at least 50 original Weibo posts around a month in total from 31 December, 2019 to 26 January, 2020. Second, their authentication type is non-institutional (e.g., individual user, etc.). Third, their regional authentication is in China, not “overseas” or “other”. 

We acquired 17,865 active Weibo users finally, then fetched all their original posts published during 13 January, 2020 to 26 January, 2020 into the two-week period for the analysis.

### 2.2. Measurement of Psychological Traits and Procedures

In this study, we used Online Ecological Recognition (OER) [20], which referred to the automatic recognition of psychological profile (e.g., anxiety, well-being, etc.) by using predictive models [17,20,21] based on ecological behavioral data from Weibo.

We employed Text Mind system developed by the Computational Cyber Psychology Laboratory at the Institute of Psychology, Chinese Academy of Sciences to extract content features [22], including Chinese word segmentation tool [17], and psychoanalytic dictionary [23]. We used the Chinese word segmentation tool to divide users’ original microblog content into words/phrases with linguistic annotations, such as verbs, nouns, adverbials, and objects [24], and then extracted psychologically meaningful categories through the simplified Chinese LIWC (Language Inquiry and Word Count) dictionary [23]. These lexical features were data sources for word frequency analysis. 

After feature extraction, we used the psychological prediction model [25] obtained from the preliminary training to predict the psychological profile of these active Weibo users. These predictive models are tools developed for online psychology research based on big data and deep learning technologies, including emotional indicators (anxiety, depression, indignation, and Oxford happiness), cognitive indicators (social risk judgment and life satisfaction), and so on. Figure 1 portrays the procedure from feature extraction to psychological indicator prediction. All the prediction models have reached a moderate correlation with questionnaire scores. The feasibility of predictive models has been repeatedly demonstrated [26,27,28].

We calculated word frequency, scores of negative emotional indicators (i.e., anxiety, depression, and indignation), positive emotional indicators (i.e., Oxford happiness), and cognitive indicators (i.e., social risk and life satisfaction) of the collected messages. We then compared the differences of psychological characteristics before and after the declaration of outbreak of COVID-19 on 20 January, 2020 through the paired sample t-test by using SPSS (Statistical Product and Service Solutions) 22, which is published by IBM (International Business Machines Corporation), New York, USA.

## 3. Results

### 3.1. Demographics

Among 17,865 active Weibo users, 25.23% were males and 77.95% were from Eastern China, which is considered the richest region in China. Ages of users who registered their birth date in their profile (*n* = 4156, 23.26%) ranged from 8 to 56 years with the median age of 33 years. The demographic profile is depicted in Table 1.

### 3.2. Linguistic Difference

In this study, we compare the LIWC categories between the week before (T-before) and after (T-after) 20 January, shown in Table 2. It contains two types of LIWC categories: words of emotions and words of concerns. Words of emotions include positive emotion (e.g., faith, contentment, and blessing), negative emotion (e.g., worry, suspicion, and jealousy), anxiety (e.g., upset, nervous, and crazy), and anger (e.g., complaint). Words of concerns include health (e.g., insomnia, doctor, and exercise), leisure (e.g., cooking, chatting, and movies), family (e.g., family and house), friend (e.g., companion and guest), money (e.g., bills, cash, and borrowing), death (e.g., burial, killing, and funeral), and religion (e.g., church, mosque, and temple), which can reflect what people are paying attention to.

After 20 January, the number of words increased in positive emotion (*t* (17,747) = −24.411, *p* < 0.001), negative emotion (*t* (17,747) = −15.273, *p* < 0.001), and anxiety (*t* (17,747) = −15.294, *p* < 0.001). Word frequency significantly increased in the category “concerns,” including health (*t* (17,747) = −72.392, *p* < 0.05), family (*t* (17,747) = −12.571, *p* < 0.001), death (*t* (17,747) = −6.707, *p* < 0.001), and religion (*t* (17,747) = −13.816, *p* < 0.001), but decreased in leisure (*t* (17,747) = 21.963, *p* < 0.001) and friend (*t* (17,747) = 6.202, *p* < 0.001).

### 3.3. Emotional Indicators

Results indicate significant differences of emotional indicators between T-before (13–19 January, 2020) and T-after (20–26 January, 2020), as shown in Table 3. After 20 January, negative emotional indicators of psychological traits increased in anxiety (*t* (17,747) = −35.962, *p* < 0.001), depression (*t* (17,747) = −10.717, *p* < 0.001), and indignation (*t* (17,747) = 5.500, *p* < 0.001), while positive emotional indicators of psychological traits decreased in Oxford happiness (*t* (17,747) = 3.120, *p* < 0.01).

### 3.4. Cognitive Indicators

We found significant differences in cognitive indicators between T-before (13–19 January, 2020) and T-after (20–26 January, 2020), as shown in Table 4. After 20 January, cognitive indicators of psychological traits increased in social risk judgement (*t* (17,747) = 3.120, *p* < 0.01), but decreased in life satisfaction (*t* (17,747) = 5.500, *p* < 0.001).

## 4. Discussion

Since the National Health Commission identified COVID-19 as a B type infectious disease officially, COVID-19 influenced the psychological states of people across China. This study collected active Weibo users’ data, and conducted sentiment analysis during 13–26 January, 2020. We used OER to acquire the psychological states, and found that Weibo users’ psychological conditions significantly changed under the outbreak of COVID-19.

The findings showed that people’s concerns by linguistic expression increased after January 20. We observe an increase in health and family, while a decrease in leisure and friend. Uncertainty of the upcoming situation causes cognitive dissonance and insecurity; this produces a feeling of mental discomfort, leading to Weibo’s activity oriented toward dissonance reduction and keeping security on health and family relationship [29]. According to the theory of BIS, people behave in a more reticent and conservative way when they feel threatened by disease [30]. Therefore, staying at home with family and reducing recreational activities seems to be a safer way to prevent illness. It also indicated that people begin to care more about their health and were more likely to seek social support from their families rather than getting together with friends, which suggested that people’ interests and attention were influenced by the restricted travel policy and self-isolation regulations from the health authorities and central government. 

Affected by COVID-19, messages related to death and religion became salient after 20 January. Reports showed severity and potential mortality of COVID-19. Research confirmed that people tended to respond to emergencies such as stress or death in the way of religion, which can comfort tense moods and bring more positive emotions [31]. That is why people prayed for the county through religion or other beliefs, leading to the phrase that appeared most frequently on the Internet at that time: God bless China.

People showed more negative emotions (anxiety, depression, and indignation) and less positive emotions (Oxford happiness) after the declaration of COVID-19, which was supported by the theory of BIS, i.e., people did generate more negative emotions for self-protection [3,4]. These results are consistent to previous studies as well, which found that public health emergencies (e.g., SARS) triggered a series of stress emotional response containing a higher level of anxiety and other negative emotions [32,33]. Meanwhile, the confirmation that COVID-19 could be passed from person to person on 20 January, which was inconsistent with previous reports, lead to quite a number of people being unsatisfied with misinformation published from provincial governments (e.g., Hubei) and ineffective regulatory actions, causing an increase in indignation. However, it’s worth noting that the word frequency of positive emotions increased after 20 January, which seemed to be inconsistent with the theory of BIS. In fact, positive emotion includes words such as faith and blessing, which are more inclined to reflect group cohesiveness rather than pure personal emotions (e.g., happiness). Researchers found that group threats (e.g., natural disasters and epidemic diseases) made groups a community of interests, resulting in more beneficial behaviors and social solidarity, which indicated higher group cohesiveness [34]. For example, lots of provinces (e.g., Sichuan Province, Shandong Province, etc.) formed medical teams to help the Hubei province, which was the worst affected area. Many people donated money and supplies to Hubei Red Cross to support the control of COVID-19.

Furthermore, social risk judgement was higher and life satisfaction was lower after the declaration of COVID-19. It is consistent with the theory of BIS, which found that when social uncertainty increased, such as unknown etiology and ambiguous route of transmission, people developed the negative cognitive assessment (e.g., higher sensitivity of risk judgment or risk perception) so that they could discover potential infection sources in time and avoid infection [2,35]. Not only that, people’s fear of potential risk and lack of controllability caused by COVID-19 brought about higher risk judgement as perceived risk theory claimed [10]. Moreover, some preventive policies and regulations in terms of travel restriction and self-isolation made the quality of life worse, reflecting in lower life satisfaction.

The following briefly foregrounds some of the study’s implications for policy makers and clinical practitioners (e.g., social workers, psychiatrists, and psychologists) plan and fight against COVID-19. For policy makers: (1) develop a consistent policy and procedure for reporting the latest confirmed cases, recent death toll, and other data about the epidemic situation. For example, the surge of cases on February 12th did not mean that the situation has been out of control, but because of the new diagnostic criteria introduced. It is important to let people understand the data properly to reduce excessive stress responses (e.g., anxiety, depression, etc.) brought on by inappropriate perception. (2) Expand public awareness of continuous progress in decision-making measures. Since indignation may come mainly from mistakes and deficiencies in preventing and controlling the epidemic, it can effectively decrease indignation if public awareness and involvement are provided. (3) Ensure the supply of medical treatment service. It is critical to set up medical service to treat the disease, and let people know how to access it conveniently. People can get help in time if they are infected. It can improve people’s sense of control over risks, thereby avoiding excessive social risk perception. (4) Provide more in-door entertainment services to address good quality of life. People may be more willing to cooperate when their living and entertainment requirements are met, such as online shopping, entertainments, etc. For clinical practitioners: (1) adjust consultant configuration rationally and cooperate with each other. Psychological consultants should grasp the epidemic information correctly and conduct science popularization during counseling. Social workers can help solve practical problems in life. These actions can improve the sense of stability and relieve anxiety and depression. (2) Deliver necessary psychosocial therapy in various ways. Considering the particularity of self-isolation, relevant hotline counseling and online consulting should be applied in practice.

Several other points should be considered when generalizing this study’s findings. First, as Weibo users are mainly young people, the results may be biased to some extent. In addition, the current analysis is based on a weekly basis, with a relatively large granularity, which has certain influences on reflecting the changing trend of social mentality in a timely manner. In further studies, we will try to expand the range of sex and age and predict psychological traits in a finer granularity. Previous studies indicated that people tended to exaggerate attitudes and prejudices, especially when they felt more vulnerable to disease transmission [36]. It inspires us to try to build a prediction model which can predict people’s attitudes and beliefs against the virus through online Weibo data for further understanding of psychological impacts of public health emergencies.

## 5. Conclusions

In this study, we compared the difference before and after 20 January on both linguistic categories and psychological profile. We found an increase in negative emotions (anxiety, depression, and indignation) and sensitivity to social risks, as well as a decrease in positive emotions (Oxford happiness) and life satisfaction after declaration of COVID-19 in China. What’s more, people show more concern for health and family, and less concern for leisure and friends. Using social media data may provide timely understanding of the impact of public health emergencies on the public’s mental health during the epidemic period.

## Figures and Tables

**Figure 1 ijerph-17-02032-f001:**
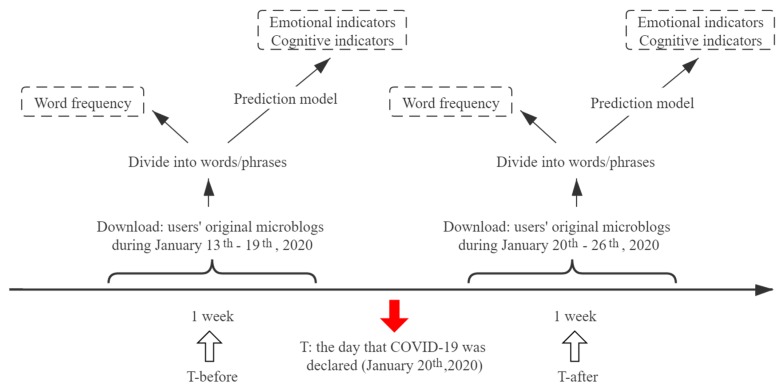
Procedures of feature extraction from online Weibo data and psychological indicator predicted by dynamic features.

**Table 1 ijerph-17-02032-t001:** Demographic characteristics of selected participants.

		*n* (%)
Gender	male	4507 (25.23)
	female	13,358 (74.77)
Age	–9	110 (0.62)
	10–19	20 (0.11)
	20–29	2035 (11.39)
	30–39	1598 (8.94)
	40–	393 (2.20)
	missing data	13,709 (76.74)
Region of location	Eastern China	13,925 (77.95)
	Central China	1644 (9.20)
	Western China	2296 (12.85)

Total		17,865 (100)

**Table 2 ijerph-17-02032-t002:** Word frequency analysis before and after 20 January.

	T-Before		T-After		t	df	*p*
	M	SD	M	SD			
**Words of emotions**							
Positive emotion	2.58	1.46	2.86	1.47	−24.411	17,747	0.000 ***
Negative emotion	0.71	0.63	0.79	0.59	−15.273	17,747	0.000 ***
Anxiety	0.09	0.17	0.12	0.17	−15.294	17,747	0.000 ***
Anger	0.19	0.26	0.19	0.23	−0.347	17,747	0.792
**Words of concerns**							
Health	0.37	0.43	0.72	0.63	−72.392	17,747	0.000 ***
Leisure	1.77	1.28	1.60	1.19	21.963	17,747	0.000 ***
Family	0.22	0.30	0.25	0.30	−12.571	17,747	0.000 ***
Friend	0.11	0.20	0.10	0.16	6.202	17,747	0.000 ***
Money	0.71	0.77	0.71	0.75	1.353	17,747	0.176
Death	0.14	0.27	0.15	0.24	−6.707	17,747	0.000 ***
Religion	0.28	0.46	0.32	0.45	−13.816	17,747	0.000 ***

T-before represents the word frequency during 13–19 January, 2020; T-after represents the word frequency during 20–26 January, 2020; M = mean; SD = standard deviation; df = degrees of freedom. *** *p* < 0.001.

**Table 3 ijerph-17-02032-t003:** Emotional indicators before and after 20 January.

	T-Before		T-After		t	df	*p*
	M	SD	M	SD			
Negative emotions							
Anxiety	11.69	4.61	12.79	4.66	−35.962	17,747	0.000 ***
Depression	14.87	4.81	15.27	5.08	−10.717	17,747	0.000 ***
Indignation	1.83	0.43	1.86	0.45	−11.415	17,747	0.000 ***
Positive emotions							
Oxford happiness	89.91	9.48	89.71	8.84	3.120	17,747	0.002 **

T-before represents the predicted emotional indicators during 13–19 January, 2020; T-after represents the predicted emotional indicators during 20–26 January, 2020; M = mean; SD = standard deviation; df = degrees of freedom. ** *p* < 0.01, *** *p* < 0.001.

**Table 4 ijerph-17-02032-t004:** Cognitive indicators before and after 20 January.

	T-Before		T-After		t	df	*p*
	M	SD	M	SD			
Social risk judgment	4.10	0.27	4.12	0.25	−8.832	17,747	0.000 ***
Life satisfaction	14.33	2.47	14.24	2.28	5.500	17,747	0.000 ***

T-before represents the predicted cognitive indicators during 13–19 January, 2020; T-after represents the predicted cognitive indicators during 20–26 January, 2020; M = mean; SD = standard deviation; df = degrees of freedom. *** *p* < 0.001.

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
