# Peer review of "The Impact of COVID-19 Epidemic Declaration on Psychological Consequences: A Study on Active Weibo Users"

_ijerph, 2020, doi:10.3390/ijerph17062032_

Round 1

Reviewer 1 Report

This study examined the impacts of public health emergencies COVID-19 on people's mental health via the Weibo data from 17,865 active Weibo users. The authors found that an increase in negative emotions such as anxiety, depression, and indignation) and a decrease in positive emotions e.g., happiness and life satisfaction after the declaration of COVID-19 in China. Overall the study is well conducted taking into consideration the special circumstances self-isolations limit due to the infectious virus and thus is more than timely and noteworthy. However, I have a few comments that need clarification.

  • In the introduction section, the authors mentioned the behavioral immune system which is considered as a rough first line of defense against disease-causing pathogens. Can the authors elaborate a little more about this? For example, which are implications for human behaviour?
  • If the authors have the data, it would be interesting to examine the attitudes and beliefs against the virus. It’s not uncommon that people tend to exaggerate attitudes and prejudices especially when they feel vulnerable to the potential transmission of infectious diseases.
  • Additionally, the authors could use the abovementioned theory to explain their results in the discussion section. One line of research has shown that the behavioural immune system people engage in more reticent and conservative forms of behavior under conditions in which they feel more vulnerable to disease transmission.
  • Also, in the discussion section I would like to see more information regarding education and prevention counselling. Is this study intended to provide recommendations and if so, should this be more clarified? Which strategies should be developed according to the authors? For, example web-based interventions, since face to face interventions seem not feasible at this time point?

Some minor comments:

 Table 1: the words age and gender should be capitalised as the words following Region of location and Total.

In conclusions, line 222: (Anxiety, Depression and Indignation) should be added a comma after the word depression.

Reviewer 2 Report

This study reports results of an online ecological recognition exercise comparing scores of emotional indicators and cognitive indicators in Weibo users messages before and after 20th January 2020 when Sars-CoV-2 epidemy was announced in China.

I have the following concerns:

1) Authors confound in the text COVID-19, the disease with Sars-Cov-2, the virus

2) Authors should acknowledge that the increase in positive emotions words after the official declaration of the epidemic does indeed not confirm - actually disproves - the BIS theory they refer to, which seems to me the main result of this study 

3) The clinical implication that the changes found by OER should prepare clinicians to provide interventions is not supported by data. This study does not prove that the changes found have any clinical significance whatsoever so any clinical reccommendations arennot justified. Actually, as it happened with structured debriefing in the past, it may well be that providing debriefing to populations hit by epidemics may lead to worse outcomes

4) People in self isolation at home could have filled in self-report online measures of anxiety and other negative and positive emotions (p.2 lines 61-62)

5) not clear if Ethical Committee approval was obtained

Round 2

Reviewer 2 Report

The revised version clearly improves the original manuscript. I still find that reccommendations in the Discussion are certainly based on common sense but very loosely based on results.

Sentence at lines 227-228 is very colloquial and needs correction
